# Genetic Stability of Driver Alterations in Primary Cutaneous Diffuse Large B-Cell Lymphoma, Leg Type and Their Relapses: A Rationale for the Use of Molecular-Based Methods for More Effective Disease Monitoring

**DOI:** 10.3390/cancers14205152

**Published:** 2022-10-20

**Authors:** Anne M. R. Schrader, Ruben A. L. de Groen, Rein Willemze, Patty M. Jansen, Koen D. Quint, Arjen H. G. Cleven, Tom van Wezel, Ronald van Eijk, Dina Ruano, J. H. (Hendrik) Veelken, Cornelis P. Tensen, Karen J. Neelis, Laurien A. Daniels, Esther Hauben, F. J. S. H. (Sherida) Woei-A-Jin, A. M. (Annemie) Busschots, Maarten H. Vermeer, Joost S. P. Vermaat

**Affiliations:** 1Department of Pathology, Leiden University Medical Center, 2333 ZA Leiden, The Netherlands; 2Department of Hematology, Leiden University Medical Center, 2333 ZA Leiden, The Netherlands; 3Department of Dermatology, Leiden University Medical Center, 2333 ZA Leiden, The Netherlands; 4Department of Pathology, University Medical Center Groningen, 9713 GZ Groningen, The Netherlands; 5Department of Radiotherapy, Leiden University Medical Center, 2333 ZA Leiden, The Netherlands; 6Department of Radiotherapy, Amsterdam University Medical Center, Location Amsterdam Medical Center, 1105 AZ Amsterdam, The Netherlands; 7Department of Pathology, University Hospitals Leuven, 3000 Leuven, Belgium; 8Department of General Medical Oncology, University Hospitals Leuven, 3000 Leuven, Belgium; 9Department of Dermatology, University Hospitals Leuven, 3000 Leuven, Belgium

**Keywords:** primary cutaneous diffuse large B-cell lymphoma, leg type, genetic stability, survival, targeted therapies, liquid biopsies, disease monitoring

## Abstract

**Simple Summary:**

Primary cutaneous diffuse large B-cell lymphoma, leg type (PCDLBCL-LT) is an aggressive cutaneous lymphoma with high response rates to initial immune-polychemotherapy but with frequent relapses and disease-related death. To study the genetic profile during the disease course, 73 samples of primary/pre-treatment and relapsed/refractory disease of 57 patients with PCDLBCL-LT were molecularly characterized, including paired analysis in 16 patients. Targeted next-generation sequencing demonstrated genetic stability of the main oncogenic driver alterations of PCDLBCL-LT, especially (hotspot) mutations in *MYD88*/*CD79B* and loss of *CDKN2A*. As nearly all patients (95%) harboured one or more of these drivers, patients could benefit from targeted therapies addressing these alterations. Additionally, genetic stability serves as a rationale for the use of molecular-based methods for disease monitoring during follow-up, improving response evaluation and early identification and intervention of disease relapses in PCDLBCL-LT patients.

**Abstract:**

Primary cutaneous diffuse large B-cell lymphoma, leg type (PCDLBCL-LT) is a rare, aggressive cutaneous lymphoma with a 5-year disease-specific survival of only ~55%. Despite high response rates to initial immune-polychemotherapy, most patients experience a disease relapse. The genetic evolution of primary and relapsed/refractory disease has only scarcely been studied in PCDLBCL-LT patients. Therefore, in this retrospective cohort study, 73 primary/pre-treatment and relapsed/refractory biopsies of 57 patients with PCDLBCL-LT were molecularly characterized with triple FISH and targeted next-generation sequencing for 52 B-cell-lymphoma-relevant genes, including paired analysis in 16 patients. In this cohort, 95% of patients harboured at least one of the three main driver alterations (mutations in *MYD88*/*CD79B* and/or *CDKN2A*-loss). In relapsed/refractory PCDLBCL-LT, these oncogenic aberrations were persistently present, demonstrating genetic stability over time. Novel alterations in relapsed disease affected mostly *CDKN2A*, *MYC*, and *PIM1*. Regarding survival, only *MYC* rearrangements and *HIST1H1E* mutations were statistically significantly associated with an inferior outcome. The stable presence of one or more of the three main driver alterations (mutated *MYD88*/*CD79B* and/or *CDKN2A*-loss) is promising for targeted therapies addressing these alterations and serves as a rationale for molecular-based disease monitoring, improving response evaluation and early identification and intervention of disease relapses in these poor-prognostic PCDLBCL-LT patients.

## 1. Introduction

Primary cutaneous diffuse large B-cell lymphoma, leg type (PCDLBCL-LT) is a rare, extranodal variant of diffuse large B-cell lymphoma (DLBCL) that is defined by skin localization without extracutaneous manifestations at the time of diagnosis. It is an aggressive type of primary cutaneous lymphoma with a 5-year disease-specific survival (DSS) of only 55% [1]. First-line treatment consists of immune-polychemotherapy (i.e., rituximab, cyclophosphamide, doxorubicin, vincristine, and prednisone). As this treatment regimen has high toxicity and is often poorly tolerated, local radiotherapy can be considered as an alternative in case of a limited number of lesions, the poor clinical condition of the patient, or in the elderly [1]. Initial therapy results in complete remission (CR) in the vast majority of PCDLBCL-LT patients; however, over two-thirds of the patients relapse within months to years [2]. Despite CR on visual inspection and positron emission tomography-computed tomography (PET-CT) scans, following large reductions in lymphoma volume by initial therapy, the high relapse rates suggest that residual disease is not properly detected and warrants more effective disease monitoring.

It is well known that the genetic profile of PCDLBCL-LT is characterized by mutations in NF-κB-associated genes, loss of *CDKN2A*, and *MYC* rearrangements [3,4,5]. Several of these alterations, including *MYD88* L265P, alterations in the B-cell receptor pathway, loss of *CDKN2A*, and *MYC* rearrangements, were reported to be associated with an inferior outcome, however, none of these observations was independently confirmed [6,7,8,9]. In addition, the genetic evolution of PCDLBCL-LT has only scarcely been studied. In order to gain more insight into the genetic profile during the disease course, we molecularly characterized a relatively large cohort of primary/pre-treatment and relapsed/refractory disease of PCDLBCL-LT patients, including paired analysis of both time points, and correlated these profiles with disease outcome.

## 2. Materials and Methods

A total of 73 skin biopsies of primary/pre-treatment and relapsed/refractory disease of a clinically well-annotated cohort of 57 patients with PCDLBCL-LT was retrospectively selected and included in this study. Cases were diagnosed with PCDLBCL-LT according to the current classification system of the World Health Organization (WHO) [10] and the WHO—European Organisation for Research and Treatment of Cancer [1] between 2000 and 2020 in the Leiden University Medical Center (LUMC), The Netherlands and the University Hospitals, Leuven (UZL), Belgium. Staging procedures, consisting of a CT scan with a bone marrow biopsy or a PET-CT scan, demonstrated no evidence of extracutaneous disease at the time of diagnosis. The study protocol was provided with a waiver of consent by the medical ethics committees of the LUMC (B19.018) and the UZL (S62445).

From all cases, formalin-fixed and paraffin-embedded (FFPE) skin biopsies of PCDLBCL-LT lesions were collected from the archives. Immunohistochemistry and fluorescence in situ hybridization (FISH) were performed in accordance to the latest WHO classification of lymphoid neoplasms [10] and reviewed by experienced pathologists (ARS, PMJ). Immunohistochemical staining was performed with the Dako Autostainer Link 48, according to the manufacturer’s recommendations, for the following antibodies: CD10 (clone 56C6 from Dako, diluted 1:20), BCL6 (clone PG-B6p from Invitrogen, diluted 1:100), MUM1 (clone MUM1p from Dako, diluted 1:100), BCL2 (clone 124 Dako, diluted 1:80), IgM (polyclonal, from Dako, diluted 1:500), and MYC (clone Y69 from ABCAM, diluted 1:100). The thresholds for stainings to be scored as positive were ≥30% for CD10, BCL6, and MUM1, and ≥50% for BCL2 and IgM. FISH with break-apart rearrangement probes was performed for *MYC* and, in case of the presence of an *MYC* rearrangement, consecutively for *BCL2* and *BCL6*, as previously described [9]. For cell-of-origin (COO) identification, more than half of the cases (*n* = 38) were analysed with Lymph2Cx on the NanoString platform, as was previously published [11].

For targeted next-generation sequencing (NGS) with the in-house developed and validated LYMFv1 panel comprising 52 B-cell lymphoma relevant genes, DNA was isolated from the tissue samples and libraries were generated and sequenced with the Ion Torrent S5-system (Thermo Fisher Scientific), as previously reported [12]. In summary, data from patients with low read count (<100) and high transition vs. transversion ratio (Ts:Tv ratio; >5), mostly due to FFPE-induced artefacts, were excluded. Next, using variant allele frequencies (VAF) of ≥10%, variant calling and annotation were performed in Geneticist Assistant NGS interpretive Workbench (SoftGenetics). All class 4 and 5 variants (determined as likely pathogenic, respectively, pathogenic) were called. In addition, all class 3 variants (with unknown significance) that were predicted as potentially pathogenic based on the high Combined Annotation Dependent Depletion-PHRED score and/or multiple prediction scores (Sift, PolyPhen, the likelihood ratio test, and MutationTaster) were reported. Given the importance of *MYD88* and *CD79B* mutations in PCDLBCL-LT and the fact that these genes contain hotspot mutations, only the *MYD88* L265P and *CD79B* Y196 mutations with VAFs <10% were also reported.

Finally, loss of *CDKN2A* was assessed by normalizing the read counts of all *CDKN2A* amplicons generated using the median value of all amplicons in that analysis. Systematic differences between amplicons were normalized using a set of 18 libraries prepared with DNA extracted from 10 non-neoplastic tonsils. *CDKN2A* was considered lost if the normalized coverage of more than 2 consecutive amplicons was below the estimated 99% confidence intervals (CIs) of these amplicons. Visualization of the results was done using the Next-Generation Sequence Expert (NGSE) Shiny app (https://git.lumc.nl/druano/NGSE, accessed on 19 September 2019).

With the identified variants in the cohort, the LymphGen 2.0 online tool (https://llmpp.nih.gov/lymphgen/lymphgendataportal.php, accessed on 13 September 2022) was implemented to ascertain the clusters as defined by Schmitz et al. [13,14]. For this analysis, the variant coding was changed to correlate with the variant coding of LymphGen 2.0, together with an annotation file for the included samples with or without information regarding the *BCL2* or *BCL6* rearrangements. Due to the lack of copy number alteration data, the A53 cluster could not be assigned within this cohort and the targeted nature of the sequencing method resulted in a reduction of the reliability and capability of LymphGen 2.0.

Statistical analyses were performed by using RStudio (R-3.6.3, including packages survival_3.3-1 and ComplexHeatmap_2.10.0). Progression-free survival (PFS), overall survival (OS), and DSS were defined as the date from the initial diagnosis to date of relapse/progression or death by lymphoma, death by any cause, and death by lymphoma, respectively. Patients were administratively censored after 5 years of follow-up or censored at the last follow-up when there was no event. The Kaplan–Meier method was used to determine median follow-up time and to construct survival curves, and they were compared with a log rank test. Hazard ratios and 95% confidence intervals (CIs) were calculated with a Cox proportional hazards model in a univariable analysis for mutational status of each gene within the panel with a Bonferroni correction for multiple testing (accounting for all tested/mutated genes).

## 3. Results

The cohort consisted of 73 samples of primary and relapsed/refractory disease of 57 patients with PCDLBCL-LT, including 16 patients with paired analysis of both primary and relapsed disease (Figure 1). An overview of the clinical characteristics is presented in Table 1.

### 3.1. Patient and Histopathological Characteristics

The cohort consisted of 30 females (53%) and 27 males (47%) with a median age at diagnosis of 78 (range, 47 to 92) years. Patients presented with single lesions in 15 cases (26%) and with multiple lesions located in one body region in 34 cases (60%; regional disease) or more body regions in eight cases (14%; multifocal disease) [2]. The legs were involved in the vast majority (47/57; 82%).

Immunohistochemically, almost all cases were positive for BCL2 (54/57; 95%) and IgM (52/55; 95%) and most cases for MUM1 (48/57; 84%). Subsets were positive for MYC (36/56; 64%), BCL6 (32/57; 56%), and CD10 (7/57; 12%). As described before, based on CD10, BCL6, and MUM1 expression, 43 cases (75%) had a non-GCB phenotype according to Hans’ algorithm, whereas six cases were classified as ABC, 16 as undeterminable/intermediate, and 14 as GCB subtype by Lymph2Cx [11].

### 3.2. Follow Up

First-line treatment consisted of local radiotherapy in 26 patients (46%), surgery in one patient (2%), and immune-polychemotherapy in 28 patients (49%), which was combined with local radiotherapy in eight patients. Two patients were not initially treated because of spontaneous remission of a single lesion in one patient and sudden cardiac death in the other patient. Despite the therapeutic heterogeneity in our cohort, there was no difference in survival between the patients treated with local radiotherapy (*n* = 26), systemic therapy (*n* = 20), and combined local radiotherapy with systemic therapy (*n* = 8) (Log rank; *p* = 0.65), nor between patients treated with local radiotherapy (*n* = 26) versus systemic treatment with or without local radiotherapy (*n* = 28) (Log rank; *p* = 0.44).

The median follow-up duration of our cohort was 3.1 (range, 0.2–16.8) years. Response to first-line treatment in our cohort was high: 53 of 55 initially treated patients (96%) had a CR. Only two patients (4%) had refractory/progressive disease. The first patient (LT_42) had progressive disease under immune-polychemotherapy and died 14 months after diagnosis. The second patient (LT_16) had refractory disease under immune-polychemotherapy combined with local radiotherapy, but achieved a prolonged CR on second-line treatment with lenalidomide and died 45 months after diagnosis from a lymphoma-unrelated cause. From the 55 patients without refractory/progressive disease, 37 patients (67%) experienced a disease relapse or died by any cause during follow-up with a median-disease-free period of 1.5 (range, 0.1–12.1) years. At time of last follow up, 20 of 57 (35%) patients were still alive, 21 patients (37%) died from their lymphoma, and 16 patients (28%) died from other causes. The 5-year DSS was 57%, the 5-year OS was 41% and the 5-year PFS was 30%.

### 3.3. Genetic Profile of the Total Cohort

In the total cohort, mutational analysis demonstrated a median number of six (range, 0 to 26) pathogenic mutations per patient (Figure 2). One case had an abundance of pathogenic variants (*n* = 26); although suggestive of FFPE-induced artefacts, this was not reflected by the low Ts:Tv ratio (3.29) in this patient.

Mutations driving NF-κB-pathway activation were detected in 48 of 57 patients (84%), mostly affecting *MYD88* (45/57; 79%), including the L265P hotspot in 39 patients (87%), followed by *CD79B* (31/57; 54%), *CARD11* (6/57; 11%), and *TNFAIP3* (5/57; 9%). In three cases, the hotspot mutations in *MYD88* and *CD79B* were detected in low VAF, i.e., LT_33 with *MYD88* L265P in 1.7% and *CD79B* Y196S in 7.5%, LT_34 with *MYD88* L265P in 5.8% and LT_42 with *MYD88* L265P in 9%, in all cases with a read count above 1,000 reads. The genes *MYD88* and *CD79B* were both mutated in 24 of 57 patients (42%). In addition, frequent mutations affected *PIM1* (21/57; 37%) and *TBL1XR1* (14/57; 25%) and there was a variety of mutations in epigenetic genes, like *KMT2D* (13/57; 23%), *BTG1* (11/57; 19%), *BTG2* (9/57; 16%), *MEF2B* (9/57; 16%), and *EP300* (7/57; 12%). Loss of *CDKN2A* and rearrangements of *MYC* were detected in (36/57; 63%) and (14/57; 25%), respectively. One of the *MYC*-rearranged patients had a double hit with *BCL6*, but none had a double hit with *BCL2* or a triple hit with *BCL6* and *BCL2*. Overall, nearly all cases (54/57; 95%) harboured one or more of the main driver alterations of PCDLBCL-LT, i.e., (hotspot) mutations in *MYD88*/*CD79B* and/or loss of *CDKN2A*.

Regarding LymphGen, the majority of PCDLBCL-LT was assigned to the MCD cluster (29/57; 51%), as based on the LymphGen 2.0 online algorithm.

### 3.4. Genetic Profile of Patients with and without Relapse/Refractory Disease

The genetic profile of patients with and without relapse/refractory disease is presented in Appendix A. The median follow-up duration was 3.1 years for patients with relapse/refractory disease (*n* = 39) and 3.6 years for patients with non-relapsed disease (*n* = 18). When comparing their genetic profiles, especially mutations and/or rearrangements in *MYC* (14/39; 36% vs. 2/18; 11%) and mutations in *TBL1XR1* (12/39; 31% vs. 2/18; 11%) and *KMT2D* (11/39; 28% vs. 2/18; 11%) were more common in patients with relapse/refractory disease than in patients without relapse/refractory disease. However, these differences were without a statistical significance (chi-square; *p* = 0.064, *p* = 0.185, and *p* = 0.191, respectively).

### 3.5. Genetic Profile of Paired Primary and Relapse/Refractory Disease

In total, 16 paired primary and relapse/refractory molecular profiles were studied in this cohort (Figure 3). The median time between the two biopsies was 1.5 (range, 0.3–11.8) years of these 16 patients. Seven patients died from lymphoma, two patients died from an unrelated cause, and seven patients were alive at the last follow-up.

Regarding the main driver alterations, the molecular profile of PCDLBCL-LT was stable between primary and relapsed disease for *MYD88* in all cases (12/12; 100%), for *CD79B* in all but one cases (9/10; 90%), and for *CDKN2A* loss for all but two cases (10/12; 83%) (Figure 3 and Appendix A). One of the patients with acquired *CDKN2A* loss in the relapse sample (LT_08) also gained a mutation in *MYC* and lost mutations in *ETV6*, *BTG2*, and *CD70*, but was stable for *MYD88* L265P at both time points. The other patient with acquired *CDKN2A* loss (LT_04) was stable for the *MYD88* and *CD79B* hotspot mutations, but also gained a mutation in *PIM1*. Regarding the other genetic alterations, 42 of 96 (44%) were identified at both time points, 26 (28%) were only detected at time of diagnosis, and 28 (30%) were new at time of disease relapse. The genes with the most discrepant variants were *CDKN2A*, *MYC*, and *PIM1* (6.25%, 17.7% and 9% of the 96 variants, respectively). Whereas most of *MYC* mutations were lost (10 lost vs. 7 gained), most of *PIM1* mutations were gained (2 lost vs. 7 gained) in the relapse/refractory samples. In particular, in patient LT_01, six pathogenic variants and a rearrangement of *MYC* were detected in the diagnostic biopsy that were not identified in the relapse sample, which attained six alternate *MYC* mutations. Another case, LT_05 gained six *CDKN2A* nucleotide variants in its relapse biopsy that were not identified in the pre-treatment sample.

Overall, in all of the 16 patients with paired analysis, at least one of the main driver alterations (*MYD88*/*CD79B*/*CDKN2A*) was genetically stable between primary and relapsed disease.

### 3.6. Survival Analysis

Given the small patient cohort, only a descriptive survival analysis was performed (Figure 4). The median PFS (17 months; Figure 4B) and median OS (44 months; Figure 4A) of our cohort correspond with other studies [8]. In addition, the presence of an *MYC* rearrangement was statistically significantly associated with an inferior survival (Log rank; *p* = 0.032; Figure 4C), as was previously reported by our group [9]. Despite other previous reports, the presence of *MYD88* L265P, mutations in the BCR pathway genes (i.e., *CD79A/B* or *CARD11*; Figure 4E), and loss of *CDKN2A* were not associated with an inferior survival [6,7,8]. Accordingly, there was no difference in survival for patients with double expressor status of MYC and BCL2, nor for patients with either a low or high mutational burden (less or equal than six mutations compared with more than six mutations, respectively; Figure 4F).

In addition, all individual genetic abnormalities were analysed with a univariable cox-regression analysis with a Bonferroni correction for multiple testing in order to determine the risk of each mutated gene (Appendix A). This analysis demonstrated that only *HIST1H1E,* a linker histone H1 protein, had an increased hazard ratio for an inferior overall and disease-specific survival (HR: 6.52, CI-95: 2.163–19.659; adjusted *p*-value = 0.044; Figure 4D). Despite this statistically significant difference, the number of patients harbouring a *HIST1H1E* mutation is low (*n* = 4) and this finding needs validation in an independent cohort.

## 4. Discussion

In this retrospective cohort study, we extensively characterized 73 biopsies of 57 patients with PCDLBCL-LT, including 16 patients with paired analysis of pre-treatment and relapse/refractory biopsies, in order to study the genetic profile during the disease course.

In general, the mutational profile of our PCDLBCL-LT cohort was characterized by NF-κB-pathway activating mutations in 84% of the patients, mostly consisting of (hotspot) mutations in *MYD88* (79%) and *CD79B* (54%). Additionally, loss of *CDKN2A* was detected in two-thirds of the patients. Nearly all patients (95%) had at least one of these three driver alterations (mutations in *MYD88*/*CD79B* and/or *CDKN2A*-loss). These results are consistent with previous studies addressing genetics in PCDLBCL-LT [3,4,5]. The genetic profile of PCDLBCL-LT overlaps with the activated B-cell (ABC) subtype of DLBCL-NOS and, in particular, other extranodal DLBCL, such as intravascular large B-cell lymphoma (IVLBCL), primary central nervous system lymphoma (PCNSL), and primary testicular lymphoma (PTL) [15,16,17]. In addition, it closely resembles the C5 signature, as reported by Chapuy et al. [18], and the MCD genotype, as reported by Schmitz et al. [13], of which the latter was identified in 51% of our patients with the LymphGen 2.0 online tool.

As examined by comparing the genetic alterations of primary and relapse/refractory disease, the three main driver alterations of PCDLBCL-LT, i.e., *MYD88* mutations, *CD79B* mutations, and loss of *CDKN2A*, were largely stable during the disease course. In addition, discrepancies in primary and relapse/refractory disease largely affected *MYC* (nucleotide variants) and *PIM1*, and, in smaller proportions, *BTG1*, *CDKN2A* (nucleotide variants), and *EP300*. These subclonal variations might be caused by ongoing SHM, as some of these genes are known targets of activation-induced deaminase (AID), the enzyme responsible for SHM [19]. These results correspond with a study from Ducharme et al., demonstrating mutations in SHM motifs in 33-100% of several affected genes, including *PIM1* [7], and expression of AID in PCDLBCL-LT patients, as reported by Dijkman et al. [7,20]. In addition, none of the *MYC* mutations in our cohort was a hotspot mutation, affecting the 57–59 amino-acid position on the protein level, as described by Cucco et al. [21].

Regarding survival, Ducharme et al. [7] identified mutations in the B-cell receptor pathway (i.e., *CD79A*/*B* or *CARD11*) as a significant indicative biomarker for relapse/progression, but this finding was not confirmed in our study (Log rank; *p* = 0.96). This discrepancy might be explained by the small cohort size for proper survival analysis (*n* = 57 in our cohort and *n* = 32 in the cohort of Ducharme et al.) and differences in the treatment regimens between both of these cohorts. In our cohort, only rearrangements of *MYC*, as was previously reported by our group [9], and mutations in *HIST1H1E* were associated with an inferior DSS/OS, but these findings require validation in independent cohorts. Whereas the relevance of *MYC* rearrangements is largely known, the relevance of *HIST1H1E* mutations remains uncertain. Recent data showed that *HIST1H1E* mutations were associated with loss of major histocompatibility complex (MHC) class I and increase in T-regulatory cell recruitment [22], and it was marked as tumour suppressor gene resulting in distinct changes in epigenetic stages [23].

When comparing the genetic profile of patients with and without relapse/refractory disease, mutations and/or rearrangements in *MYC* (36% vs. 11%) and mutations in *TBL1XR1* (31% vs. 11%) and *KMT2D* (28% vs. 11%) were more common in patients with relapse/refractory disease than in patients without relapse/refractory disease, however, these differences were without a statistical significance.

Despite that the group size for a rare disease, such as PCDLBCL-LT, is relatively large, analysis of our data was limited by the reasonably small sample size, especially reducing the validity of our survival analysis. In addition, in five patients, genetic analysis of only relapse/refractory biopsies were included, which could have been affected by initial treatment. However, as the results of our study demonstrated genetic stability of the three main driver alterations in PCDLBCL-LT, we reasoned that these cases could be included in the survival analysis. Another limitation is the targeted approach of our molecular analysis. Therefore, our results lack the width of more comprehensive sequencing methods, such as whole genome/exome sequencing. Amongst others, this may have impaired the molecular classification with the LymphGen 2.0 online tool, as for example, our data lacked the status of copy number alterations and several genes that are required for classification. Nonetheless, the clustering-classification precision of MCD with our panel was 84% (90% sensitivity and 96% specificity). Finally, our cohort was treated heterogeneously, although the patients’ OS did not differ between the different therapies (Log rank; *p* = 0.5).

Despite that nearly all PCDLBCL-LT patients in our cohort reached complete remission after initial therapy (96%), two-thirds still experienced a disease relapse. This high relapse rate suggests that residual disease is not properly detected by our currently standard methods of physical examination and PET-CT scanning [24], implying the need for more accurate assessment of minimal residual disease (MRD) in PCDLBCL-LT patients. The genetic stability of the main driver alterations in PCDLBCL-LT patients, as demonstrated in this study, may serve as a rationale for molecular-based methods of MRD assessment. One such molecular method is analysis of tumour-specific mutations in cell-free tumour DNA obtained from liquid biopsies with digital droplet PCR (ddPCR). This technique is especially suitable for detection of hotspot alterations, such as *MYD88* L265P and *CD79B* Y196, and already allowed for non-invasive real-time monitoring of disease progression in systemic DLBCL patients [24,25,26]. Assessment of the molecular MRD can be important for monitoring disease activity, not only to determine treatment efficacy, but also as an early marker for relapsed disease when a patient is radiologically and clinically in CR but the molecular MRD marker is positive. The presence or absence of molecular detectable disease may, subsequently, direct therapeutic strategies, for example by the development of stopping rules for tyrosine kinase inhibitors when patients achieve enduring deep molecular remissions. However, these concepts will still need to be investigated in future clinical studies to determine its actual therapeutic value. In addition to molecular MRD, the stable genetic profile also serves as a rationale for targeted therapies in second-line treatment of patients with relapsed disease, such as with Bruton’s tyrosine kinase (BTK)-inhibitor ibrutinib [27]. So far, several case reports have demonstrated prolonged responses to ibrutinib in PCDLBCL-LT [28,29,30], but clinical trials are still lacking. A potential drawback of ibrutinib treatment is the development of resistance mechanisms, for example by the development of mutations in *BTK* itself or in downstream components, such as *CARD11*, as was described in PDCLBCL-LT by Fox et al. [31].

## 5. Conclusions

Our study demonstrates that the three main oncogenic drivers of PCDLBCL-LT, i.e., mutations in *MYD88* and *CD79B* and loss of *CDKN2A*, are persistently present in primary and relapse/refractory disease. Given the high relapse rates of these patients despite reaching CR on initial therapy, our results provide a rationale for the use of targeted therapies addressing these alterations and molecular-based methods in disease monitoring of PCDLBCL-LT patients.

## Figures and Tables

**Figure 1 cancers-14-05152-f001:**
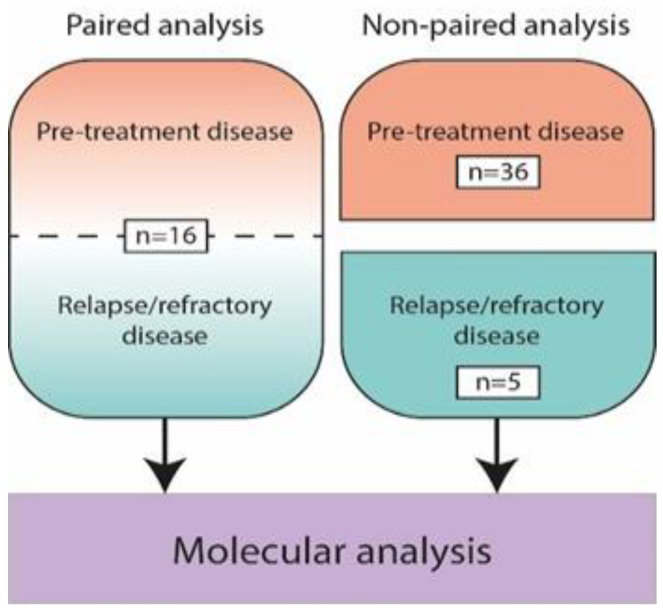
Overview of 73 included biopsy samples from pre-treatment and relapse/refractory disease of 57 patients with primary cutaneous diffuse large B-cell lymphoma, leg type, including paired analysis of both time points in 16 patients.

**Figure 2 cancers-14-05152-f002:**
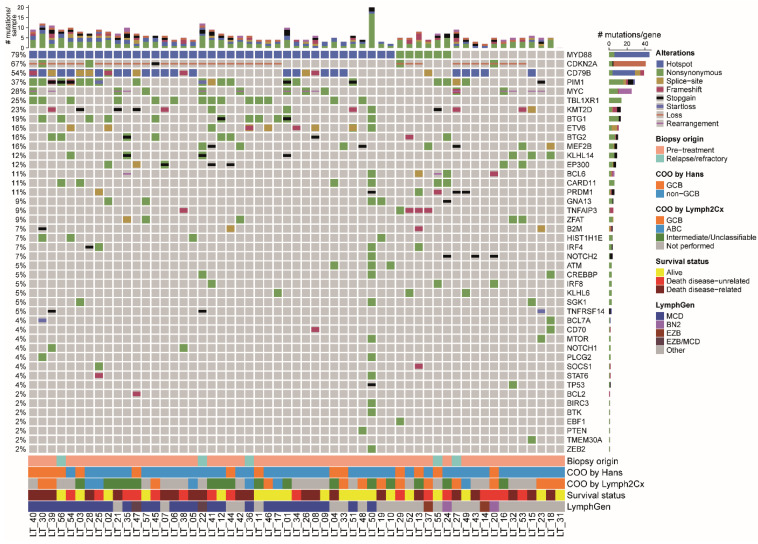
Molecular oncoprintplot of the total cohort of 57 patients with primary cutaneous diffuse large B-cell lymphoma, leg type. The type of alteration of each gene (rows) is visualized per patient (columns). The top bar chart shows the number of mutation types per patients and the right bar chart shows the number of alterations per gene. Biopsy origin (pre-treatment or relapsed/refractory disease), cell-of-origin status as determined by the Hans’ algorithm with immunohistochemistry or Lymph2Cx with NanoString, survival status, and estimated LymphGen classification are presented at the bottom. The most frequent alterations were mutations in *MYD88* and *CD79B*, especially the hotspots L265P and Y196, respectively, and loss of *CDKN2A*. In 95% of patients, at least one of these three abnormalities was present.

**Figure 3 cancers-14-05152-f003:**
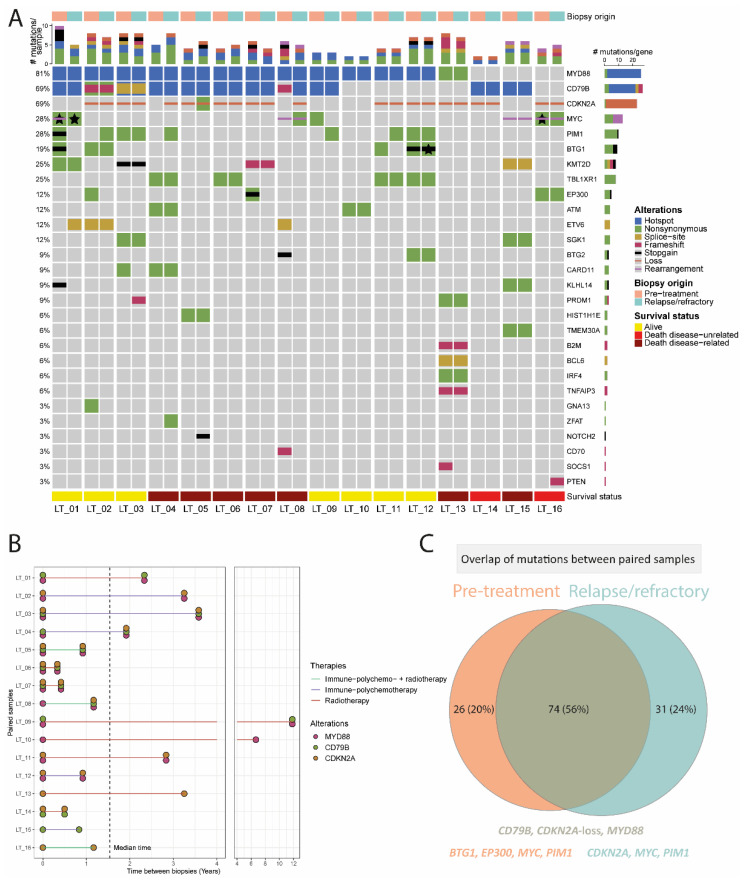
Paired molecular analysis of 16 patients with primary cutaneous diffuse large B-cell, leg type. (**A**) Side-by-side oncoprint of the molecular profile of pre-treatment and relapse/refractory samples. Each patient harboured at least one (hotspot) alteration in *MYD88*, *CD79B*, or *CDKN2A*. One patient gained a loss of *CDKN2A* and another patient gained a *CD79B* hotspot mutation and loss of *CDKN2A*. The stars indicate the presence of the same mutation type (e.g., nonsynonymous) in the primary and relapse samples but affecting different parts of the gene. (**B**) Time between biopsies of the paired samples with a median time of 1.5 (range, 1.3–11.8) years. Half of the patients received immuno-polychemotherapy (three patients with and five patients without local radiotherapy, including one patient who received R-CEOP instead of R-CHOP). The remaining patients were treated with local radiotherapy without systemic agents. Only two out of the 16 patients had a shift in the three major drivers. (**C**) Overview of the unique alterations identified in primary and relapse cases and the overlap between both time points. Most of the genetic alterations (74/131; 56%) were stable over time, especially mutations in *MYD88*/*CD79B* and loss of *CDKN2A*. Most of the mutations that were gained (31/131; 24%) affected *CDKN2A*, *MYC*, and *PIM1*, and most of the mutations that were lost (26/131; 20%) affected *BTG1*, *EP300*, *MYC,* and *PIM1*.

**Figure 4 cancers-14-05152-f004:**
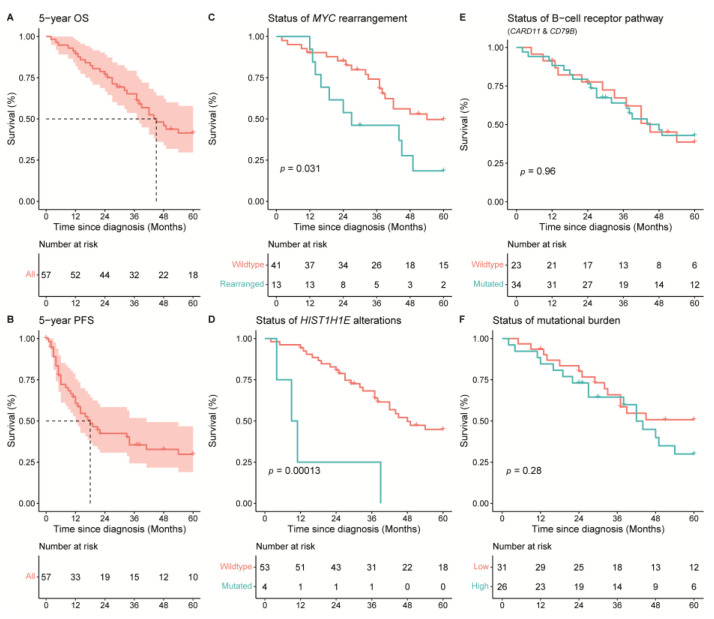
Overall survival (OS) and progression-free survival (PFS) of 57 patients with PCDLBCL-LT. The median OS of the cohort was 44 months. (**A**) The median PFS of the cohort was 17 months. (**B**) Rearrangements of *MYC* (**C**) and mutations in *HIST1H1E* (**D**) were associated with an inferior survival (Log rank; *p* = 0.001 and *p* = 0.00013, respectively). There were no statistically significant differences in OS for (**E**) patients with or without mutations in the B-cell receptor pathway (*CD79A/B* or *CARD11*) and (**F**) patients with a high (>6 mutations) or low (≤6 mutations) mutational burden.

**Table 1 cancers-14-05152-t001:** Clinical and immunophenotypic characteristics of 57 patients with primary cutaneous diffuse large B-cell lymphoma, leg type.

	All Patients (*n* = 57)
**Sex**, *n* (%)—Male	27 (47)
**Median age at diagnosis**, years (range)	78 (47–92)
**Disease localization at time of diagnosis**, *n* (%)	
Legs	47 (82)
Other skin sites ^1^	10 (18)
**Disease extension at time of diagnosis**, *n* (%)	
Single lesion	15 (26)
Regional	34 (60)
Multifocal	8 (14)
**First-line therapy**, *n* (%)	
Local treatment ^2^	27 (47)
Systemic treatment ^3^	20 (35)
Combined (local + systemic) treatment ^4^	8 (14)
No treatment ^5^	2 (4)
**Response to first-line therapy**, *n* (%)	
Complete remission	55 (96)
Refractory/progressive disease	2 (4)
**Median follow-up duration**, years (range)	3.1 (0.2–16.8)
**Occurrence of disease relapses**, *n* (%) Median disease-free period, years (range)	37 (67) 1.5 (0.1–12.1)
**Status at last follow-up**, *n* (%)	
Alive	20 (35)
Died of lymphoma	21 (37)
Died of other cause	16 (28)
**Survival**, %	
5-year disease-specific survival 5-year overall survival	57 41
5-year progression-free survival	30
**Immunophenotype**, *n* (%)	
CD10 expression	7 (12)
BCL6 expression	32 (56)
MUM1 expression	48 (84)
BCL2 expression	54 (95)
IgM expression	52 (95) ^6^
MYC expression	36 (64) ^7^

^1^ Other skin sides included the arms in four patients, the trunk in three patients, the head-and-neck region in two patients, and the genital aera in one patient. ^2^ Local treatment consisted of radiotherapy in 26 patients and surgical excision in one patient. ^3^ Systemic treatment consisted of a combination of rituximab plus cyclophosphamide, doxorubicin, vincristine, and prednisone (R-CHOP) in 16 patients, R-CHOP with etoposide instead of doxorubicin (R-CEOP) in one patient, and CHOP in three patients. ^4^ Combined treatments consisted of local radiotherapy combined with R-CHOP in six patients, local radiotherapy combined with vincristine and prednisone (OP) in one patient, and local radiotherapy combined with R-CEOP in one patient. ^5^ No treatment was given in two patients due to spontaneous remission of a single lesion in one patient and sudden cardiac death in the other patient. ^6^ Data is missing in two cases. ^7^ Data is missing in one case.

## Data Availability

The raw data presented in this study are openly available in the sequencing read archive (SRA) at https://www.ncbi.nlm.nih.gov/sra/PRJNA629460 (accessed on 16 August 2022), reference number PRJNA629460.

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
