# Peer review of "Genetic Stability of Driver Alterations in Primary Cutaneous Diffuse Large B-Cell Lymphoma, Leg Type and Their Relapses: A Rationale for the Use of Molecular-Based Methods for More Effective Disease Monitoring"

_cancers, 2022, doi:10.3390/cancers14205152_

Round 1
Reviewer 1 Report
The authors present a series of patients with primary cutaneous DLBCL, leg type with NGS sequencing performed using a small custom capture 52 gene panel. Samples from 57 patients were analyzed including 16 paired biopsy samples. The authors present analysis of the results of the NGS including the matched, paired specimens, and correlate with clinical outcome.
Major:
The method used to detect CDKN2A loss is not described in the methods section and it is unclear how this analysis was performed.
The authors used a VAF frequency of >/= 10% as the threshold for including mutations within the custom capture panel, but then create an exception for MYD88 and CD79B allowing VAF < 10%. The reason for this inconsistency in analysis is explained that these are key mutations in the disease and with known hot-spot mutations. The threshold VAF for MYD88 and CD79B should be specifically listed and the justification to use different thresholds for MYD88 and CD79B versus other genes in the panel is not satisfactory.
The authors state that targeted therapy specific to MYD88/CD79B mutations and CDKN2A loss should be considered for PCDLBCL-LT based upon the high frequency of alterations. How these might be targeted and published literature on the effectiveness of targeting this genetic alterations should be provided or the authors should soften the language about the targetability of these genetic alterations.
The novelty of this analysis is modest due to the lack of data regarding structural alterations and the relatively small size of the custom capture panel limiting the ability to assign cases according to the LymphGen 2.0 classifier. The addition of a more extensive panel and/or copy number alteration determination by low pass whole exome sequencing or other methods would significantly increase the novelty and relevance of this study.
There is overlap with previously published work by the same group (which is acknowledged by the authors) regarding the prognostic significant of MYC rearrangement assessed by FISH with outcomes.
Author Response
Reviewer 1
The authors present a series of patients with primary cutaneous DLBCL, leg type with NGS sequencing performed using a small custom capture 52 gene panel. Samples from 57 patients were analyzed including 16 paired biopsy samples. The authors present analysis of the results of the NGS including the matched, paired specimens, and correlate with clinical outcome.
Major:
Comment: The method used to detect CDKN2A loss is not described in the methods section and it is unclear how this analysis was performed.
Reply: As the reviewer correctly noted, our manuscript unfortunately lacked the description about the method of determining the loss of CDKN2A. As such, we have added the following section to the manuscript: (Page 3, Line 118-125): “Finally, loss of CDKN2A was assessed by normalizing the read counts of all CDKN2A amplicons generated using the median value of all amplicons in that analysis. Systematic differences between amplicons were normalized using a set of 18 libraries prepared with DNA extracted from 10 non-neoplastic tonsils. CDKN2A was considered lost if the normalized coverage of more than 2 consecutive amplicons was below the estimated 99% confidence intervals (CIs) of these amplicons. Visualization of the results was done using the Next-Generation Sequence Expert (NGSE) Shiny app (https://git.lumc.nl/druano/NGSE).”
Comment: The authors used a VAF frequency of >/= 10% as the threshold for including mutations within the custom capture panel, but then create an exception for MYD88 and CD79B allowing VAF < 10%. The reason for this inconsistency in analysis is explained that these are key mutations in the disease and with known hot-spot mutations. The threshold VAF for MYD88 and CD79B should be specifically listed and the justification to use different thresholds for MYD88 and CD79B versus other genes in the panel is not satisfactory.
Reply: The variant allele frequency threshold of 10% was set to prevent the influence of potential artefacts (induced by the use of FFPE as storage medium) on the analysis, especially since PCDLCBL-LT is a rare disease and we included samples from the year 2000 and onwards (with an increase in these artefacts in older samples). Because MYD88 L265P and CD79B Y196 are well known hotspots mutations in DLBCL and highly prevalent in PCDLBCL-LT, we assumed that the potential for the presence of such specific alterations to be a FFPE-induced artefact rather than a somatic mutation was extremely low. Accordingly, we decided to include these hotspot mutations with any allele frequency. In total, three MYD88 L265P mutations were detected with 1.7, 5.8, and 9% and one CD79B Y196 mutation with 7.5%. To improve the clarity of the manuscript, we have added/altered the following sentences to/in the manuscript (Page 3, Line 116): “Given the importance of MYD88 and CD79B mutations in PCDLBCL-LT and the fact that these genes contain hotspot mutations, only the MYD88 L265P and CD79B Y196 mutations with VAFs <10% were also reported.” and (Page 6, Line 210-214) “In three cases, the hotspot mutations in MYD88 and CD79B were detected in low variant allele frequency, i.e., LT_33 with MYD88 L265P in 1.7% and CD79B Y196S in 7.5%, LT_34 with MYD88 L265P in 5.8% and LT_42 with MYD88 L265P in 9%, in all cases with a read count above 1,000 reads. ”
Comment: The authors state that targeted therapy specific to MYD88/CD79B mutations and CDKN2A loss should be considered for PCDLBCL-LT based upon the high frequency of alterations. How these might be targeted and published literature on the effectiveness of targeting this genetic alterations should be provided or the authors should soften the language about the targetability of these genetic alterations.
Reply: As rightly noted by the reviewer, we stated that these molecular markers could be targeted with specific therapy regiments, but we did not include published literature supporting these targeted therapy possibilities. This lack was also noticed by reviewer 2 and we have now included the following in the discussion section of the manuscript (Page 12, lines 423-430): “In addition to disease monitoring, the stable genetic profile of MCD cluster also serves as a rationale for targeted therapies in second-line treatment of patients with relapsed disease, such as with the Bruton’s tyrosine kinase (BTK) inhibitor ibrutinib [Wilson et al. Cancer cell 2021]. So far, several case reports have demonstrated prolonged responses to ibrutinib in PCDLBCL-LT [Deng et al. Haematologica 2017; Gupta et al. Rare tumors 2015; Pang et al. Ann Hematol 2019], but clinical trials are still lacking. A potential drawback of ibrutinib treatment is the development of resistance mechanisms, for example by development of mutations in BTK itself or in downstream components, such as CARD11, as was described in PDCLBCL-LT by Fox et al. [Int J Mol Sci 2018].”
Comment: The novelty of this analysis is modest due to the lack of data regarding structural alterations and the relatively small size of the custom capture panel limiting the ability to assign cases according to the LymphGen 2.0 classifier. The addition of a more extensive panel and/or copy number alteration determination by low pass whole exome sequencing or other methods would significantly increase the novelty and relevance of this study.
Reply: We agree with the reviewer that the custom capture panel used in our study has limitations with regard to the LymphGen 2.0 classifier and that the results of our study would benefit from a more extensive panel and/or copy number alteration determination. Nonetheless, with our limited panel and exclusion of mostly BCL2/BCL6-rearrangement status and copy number alteration data, we still attained an estimated clustering-classification precision of 84% for MCD (90% sensitivity and 96% specificity), as reported in the manuscript (Page 11, lines 386-387): “Nonetheless, the clustering-classification precision of MCD with our panel was 84% (90% sensitivity and 96% specificity).”
Comment: There is overlap with previously published work by the same group (which is acknowledged by the authors) regarding the prognostic significant of MYC rearrangement assessed by FISH with outcomes.
Reply: The reviewer is correct in its assessment that there is (little) overlap with our previous work. However, the size of the cohort has increased and the focus of the study has shifted to a custom-tailored panel rather than MYC translocation status only. Nonetheless, it remains important to verify previous findings to further confirm their validity, as we stated in the manuscript (Page 11, lines 361-362): “…, but these findings require validation in independent cohorts”.
Reviewer 2 Report
General comments :
Shrader et al report a pooled cohort of paired and non paired cases of Leg-type DLBCL genetically explored. The main interest of this work resides in the paired samples and the interest of additional non paired cases is questionable.
Datas are clearly presented despite the lack of some information (see infra) that should be added or justified. Especially, loss or gain of mutations described in fig 3C should be highlighted case by case in Fig 3A as they do not appear clearly.
Discussion may be improved with comments on the relevance of these data showing genetic stability upon treatment in regards with interest of iBtk treatment strategy in non GCB DLBCL.
Typo :
Page 2 Line 68 :
In de B-cell receptor
Page 4 :
Table 1 :
Presence or absence of extra-cutaneous extension should be precised in addition in the disease extension part.
BCL2, CL6 and MYC expression should be described precisely with the used threshold to define positivity.
Fig2 : the grey color of Lymph2x status is not legended
Page 8 : Line 240 : “Time between biopsies of the paired-patients, whereas the median time was 1.5 (range 1.3-11.8) years. Half of the patients received immuno-polychemotherapy (with or without local radiotherapy).” It should be interesting to know which treatement the other patients received? Alkylating agents? Radiotherapy? Targeted therapy? If mentioned it is not clearly evident for reader when comparing with Table 1.
Page 8 : Paragraph on oncogenic drivers beginning line 247 :
One missing information is the variant allele frequencies and their stability in each sample between diagnosis and relapse. A summary table should be helpful.
Page 8 line 284 : HIST1H1E, is not a histone modifier, it is a gene coding the linker histone H1E. See Yusufova et al, Nature 2020.
CD10 expression is quite surprising in 7 patients : do the authors specifically excluded primary cutaneous follicular lymphoma. Do these patients exhibit different genetic profiles? If yes, these samples may be excluded.
Figure 3 : There is discrepancies between data Fig 3A and 3C. Fig 3C suggest that there is gains in mutations of MYC, PIM1, BTG1 and EP300. These data are not seen in the 3A figure (with mainly disappearance of mutations in some patients). Do the authors comment these differences and correct it?
Description of MYC mutation patterns should be interesting. Do they rely on hotspot sites with impact on proteic expression like T58 (see Cucco et al)? If yes, MYC expression by IHC should be interesting to add on the paired cases considering the appearance of these mutations in this dataset.
One major issue is the limited sensitivity of NGS sequencing. This issue impairs the capacity of detection of sub-clonal mutations already present at diagnosis. It should be interesting to specifically focus on presence or absence of mutations detected at relapse on diagnostic paired samples with a threshold of 1 or 2% as applied for the hotspot mutations.
Discussion :
The discussion should include a part related to targeted therapies. As non GC DLBCL (and notably MCD cluster) is reported to be sensitive on Btk inhibitors (PMID: 34739844), this strategy is quite interesting in leg-type DLBCL. Here, the genetic stability of these lymphomas, and notably on MYD88 and CD79B mutations suggesting a strong driver impact may represent an interesting point in favor of these treatments.
Page 9 : Giving the high rate of MYD88 and CD79B, it is quite surprising to only have a proportion of 51% of samples classified as MCD.
Page 9 : line 314, this paragraph specify that differences between diagnosis and relapse are largely due to SHM but authors only refers to literature to justify this assertion. As these data are not clearly reported in the results section authors would either temper this sentence or apply AID signature testing on sequencing data and add this results before affirming the AID implication in these process.
Page 10 : line 322 : Authors should argument their discrepancies with Ducharme et al on the absence of prognostic impact found of MYD88 and CD79B mutations in contrary to this paper which is probably due to the high rate of these mutations in the current serie.
Page 10 : Paragraph beginning line 349. This paragraph is of interest but this example on one patient is not really meaningfull isolated. It should not be mentioned if other cases cannot be added as example.
Author Response
Reviewer 2
General comments :
Schrader et al report a pooled cohort of paired and non paired cases of Leg-type DLBCL genetically explored. The main interest of this work resides in the paired samples and the interest of additional non paired cases is questionable.
Datas are clearly presented despite the lack of some information (see infra) that should be added or justified. Especially, loss or gain of mutations described in fig 3C should be highlighted case by case in Fig 3A as they do not appear clearly.
Discussion may be improved with comments on the relevance of these data showing genetic stability upon treatment in regards with interest of iBtk treatment strategy in non GCB DLBCL.
Comment: Typo Page 2 Line 68: In de B-cell receptor
Reply: We changed this in the manuscript.
Comment: Page 4, Table 1:
Presence or absence of extra-cutaneous extension should be precised in addition in the disease extension part.
Reply: At time of diagnosis, staging procedures demonstrated no evidence of extracutaneous disease. Therefore, disease extension only applies to the extension in the skin. We added “at time of diagnosis” to the “disease localization” and “disease extension” part of table 1. Also, we added the following sentence to the materials and methods section (Page 2, line 83-85): “Staging procedures, consisting of a computed tomography (CT) scan with a bone marrow biopsy or a positron emission tomography - CT scan, demonstrated no evidence of extracutaneous disease at time of diagnosis.”
Comment: BCL2, BCL6 and MYC expression should be described precisely with the used threshold to define positivity.
Reply: As requested, the following sentence was added to the materials and methods section of the manuscript (Page 3, lines 97-98): “The thresholds for a staining to be scored as positive were ≥30% for CD10, BCL6, and MUM1, and ≥50% for BCL2 and IgM.”
Comment: Fig2 : the grey color of Lymph2x status is not legended.
Reply: The grey color indicates that this analysis was not performed. We added this to the figure legend.
Comment: Page 8: Line 240 : “Time between biopsies of the paired-patients, whereas the median time was 1.5 (range 1.3-11.8) years. Half of the patients received immuno-polychemotherapy (with or without local radiotherapy).” It should be interesting to know which treatment the other patients received? Alkylating agents? Radiotherapy? Targeted therapy? If mentioned it is not clearly evident for reader when comparing with Table 1.
Reply: We changed the sentence and added the treatment strategies to the legend of Figure 3 (Page 8, line 254-256): “Half of the patients received immuno-polychemotherapy (three patients with and five patients without local radiotherapy, including one patient who received R-CEOP instead of R-CHOP). The remaining patients were treated with local radiotherapy without systemic agent.”
Comment: Page 8: Paragraph on oncogenic drivers beginning line 247:
One missing information is the variant allele frequencies and their stability in each sample between diagnosis and relapse. A summary table should be helpful.
Reply: To accommodate the concerns of the reviewer with regards to this comment as well as the comment for mutations below our threshold of 10% variant allele frequency, we included a supplemental figure 2. This figure lists an overview of all mutations with a minimum of 10% variant allele frequency with a read count of at least 100 reads, or ‘failed’ variants (<10%) that are paired with a successful variant (>10%). Within this figure, each pathogenically-called variant is represented by a dot, and colored according to its representing gene. These variants are plotted based on the allele frequency (y-axis) and presence in either primary or relapse/refractory biopsy (x-axis) of each individual patient. Additionally, the paired variants are connected with an identically colored line.
Comment: Page 8 line 284 : HIST1H1E, is not a histone modifier, it is a gene coding the linker histone H1E. See Yusufova et al, Nature 2020.
Reply: We thank the reviewer for this sharp observation and we changed this in the manuscript. We also referred to the publication by Yusufova et al. in the discussion of our manuscript (Page 11, line 363-366): ”Recent data showed that HIST1H1E mutations were associated with loss of major histocompatibility complex (MHC) class I and increase in T-regulatory cell recruitment [22], and it was marked as tumour suppressor gene resulting in distinct changes in epigenetic stages [23].”
Comment: CD10 expression is quite surprising in 7 patients: do the authors specifically excluded primary cutaneous follicular lymphoma. Do these patients exhibit different genetic profiles? If yes, these samples may be excluded.
Reply: We agree with the reviewer that CD10 expression is uncommon in PCDLBCL-LT, but to our best knowledge it has never been used as an exclusion criterion for diagnosis of PCLDBCL-LT. As we also described in our previous publication regarding COO classification in PCDLBCL-LT (Schrader et al. Virchows Archiv 2022), the CD10-positive cases in our cohort all showed the characteristic features of PCDLBCL-LT, including presentation on the leg(s), round-cell morphology, lack of (residual) follicular dendritic cell-networks, expression of BCL2 and MUM1 in all but one cases, IgM expression in all cases, and MYD88 mutations in all cases. Based on these characteristics, the cases in our cohort cannot be classified as primary cutaneous follicle center lymphoma and, therefore, we anticipate that there is no reason to use CD10 expression for exclusion as PCDLBCL-LT.
Comment: Figure 3: There is discrepancies between data Fig 3A and 3C. Fig 3C suggest that there is gains in mutations of MYC, PIM1, BTG1 and EP300. These data are not seen in the 3A figure (with mainly disappearance of mutations in some patients). Do the authors comment these differences and correct it?
Reply: As the reviewer correctly noted, there was a discrepancy between Figure 3A and 3C: the text annotation of figure 3C of CDKN2A and BTG1 and EP300 were unfortunately wrongfully placed. We thank the reviewer for the possibility to correct this error. Additionally, to indicate differences in mutations that are now visualized as the same mutation types (and thereby giving the impression that these mutations are identical) we added stars to Figure 3A in case the mutations were of the same type (e.g. nonsynonymous) but affected different parts of the gene.
Comment: Description of MYC mutation patterns should be interesting. Do they rely on hotspot sites with impact on proteic expression like T58 (see Cucco et al)? If yes, MYC expression by IHC should be interesting to add on the paired cases considering the appearance of these mutations in this dataset.
Reply: We agree with the reviewer that the mutational pattern of MYC mutations could be interesting. The reported MYC mutational hotspots spanning the 57-59 amino-acid position on the protein level, as described by Cucco et al., however, were not detected in our cohort. We have added this to the discussion section of our manuscript (Page 11, line 352-354): “In addition, none of the MYC mutations in our cohort was a hotspot mutation, affecting the 57-59 amino-acid position on the protein level, as described by Cucco et al. [21].”
Comment: One major issue is the limited sensitivity of NGS sequencing. This issue impairs the capacity of detection of sub-clonal mutations already present at diagnosis. It should be interesting to specifically focus on presence or absence of mutations detected at relapse on diagnostic paired samples with a threshold of 1 or 2% as applied for the hotspot mutations.
Reply: Indeed, there is a limiting factor in the sensitivity of next-generation sequencing techniques. However, the intended use of targeted NGS is to improve this sensitivity compared with other NGS techniques, like whole-genome or whole-exome sequencing. Nonetheless, we aimed to improve our manuscript by adding supplemental figure 2 (as was also addressed in a previous comment).
Comment: Discussion:
The discussion should include a part related to targeted therapies. As non GC DLBCL (and notably MCD cluster) is reported to be sensitive on Btk inhibitors (PMID: 34739844), this strategy is quite interesting in leg-type DLBCL. Here, the genetic stability of these lymphomas, and notably on MYD88 and CD79B mutations suggesting a strong driver impact may represent an interesting point in favor of these treatments.
Reply: We added the following lines to the discussion section of the manuscript (Page 12, lines 419-426): “In addition to disease monitoring, the stable genetic profile of MCD cluster also serves as a rationale for targeted therapies in second-line treatment of patients with relapsed disease, such as with the Bruton’s tyrosine kinase (BTK) inhibitor ibrutinib [Wilson et al. Cancer cell 2021]. So far, several case reports have demonstrated prolonged responses to ibrutinib in PCDLBCL-LT [Deng et al. Haematologica 2017; Gupta et al. Rare tumors 2015; Pang et al. Ann Hematol 2019], but clinical trials are still lacking. A potential drawback of ibrutinib treatment is the development of resistance mechanisms, for example by development of mutations in BTK itself or in downstream components, such as CARD11, as was described in PDCLBCL-LT by Fox et al. [Int J Mol Sci 2018].”
Comment: Page 9: Giving the high rate of MYD88 and CD79B, it is quite surprising to only have a proportion of 51% of samples classified as MCD.
Reply: We agree with the reviewer that this is indeed lower than expected, however, our results are based on an approximation of the LymphGen algorithm as we lack extensive copy number analysis and the rearrangement status of BCL2 and BCL6 in case no MYC rearrangement was detected in our cohort. Therefore, the adequacy of the algorithm to define MCD clusters is reduced with 15%. Looking at all cases with either a CD79B or MYD88 mutation, about 66% is classified as MCD. Additionally, the MYD88 L265P mutation and CD79B mutations are not unique to the MCD clusters, as these mutations can be found within ~15% of the BN2 or A53 clusters. As such, more than (both) these genes are necessary to define the MCD cluster. Lastly, in the publication discussing the LymphGen algorithm, approximately 38% of cases were not classified in any of the clusters with the availability of more complete molecular data.
Comment: Page 9: line 314, this paragraph specify that differences between diagnosis and relapse are largely due to SHM but authors only refers to literature to justify this assertion. As these data are not clearly reported in the results section authors would either temper this sentence or apply AID signature testing on sequencing data and add this results before affirming the AID implication in these process.
Reply: As we cannot apply proper AID signature testing (based on the targeted approach of our analysis), we changed this sentence into the following: “These subclonal variations might be caused by ongoing SHM, as some of these genes are known targets of activation-induced deaminase (AID), the enzyme responsible for SHM [19].”
Comment: Page 10: line 322: Authors should argument their discrepancies with Ducharme et al on the absence of prognostic impact found of MYD88 and CD79B mutations in contrary to this paper which is probably due to the high rate of these mutations in the current serie.
Reply: Ducharme et al. described an association between mutations in the B-cell receptor genes CD79A/B and/or CARD11 with an inferior outcome (not MYD88). In our cohort, 34 patients complied with these criteria, but no difference in survival was demonstrated (Log-rank p=0.96). We added this specific survival curve to Figure 4E and removed the curve for mutations in NF-kB-associated genes, as indeed the vast majority of patients fell into the mutated category. This discrepancy between our results might be caused by the small cohorts for proper survival analyses (Ducharme 32 patients and our cohort 57 patients) and the difference in treatment regimens between both cohorts. As such, we have added this to the manuscript (Page 11, Lines 357-359).
Comment: Page 10: Paragraph beginning line 349. This paragraph is of interest but this example on one patient is not really meaningful isolated. It should not be mentioned if other cases cannot be added as example.
Reply: One case is indeed not meaningful. We removed the case description from the paragraph.
Reviewer 3 Report
Thank you very much for providing an opportunity to review the article titled “ Genetic Stability of Driver Alterations in Primary Cutaneous Diffuse Large B-cell Lymphoma, Leg Type and Their Relapses a Rationale for the Use of Molecular-Based Methods for More Effective Disease Monitoring” by Anne M.R. Schrader and co-authors.
In this manuscript Anne M.R. Schrader and the co-authors have performed a systematic retrospective investigation on the genetic evolution of primary and relapsed/refractory primary cutaneous diffuse large B-cell lymphoma, leg type (PCDLBCL-LT) using formalin-fixed paraffin-embedded skin biopsies from 57 patients. They have used targeted next-generation sequencing technique to identify mutational status of 52 B-cell lymphoma relevant genes. In addition, patient samples were characterized by using immunohistochemistry and triple FISH for BCL2, MYC and BCL6.
The authors have reported the clinical and immunophenotypic characteristics of these patient samples. The mutational analysis identified several mutations in the key driver genes as well as other genes relevant for the B-cell lymphoma. Overall, majority of the cases had at least one or more main driver mutations of PCDLBCL-LT.
Mutational profiling of the paired samples from 16 primary and relapsed/refractory patient samples also showed at least one hotspot mutation in either MYD88, CD79B or CDKN2A. However, the mutational landscape did not change significantly over time in these paired samples.
Major comments:
The study was conducted in a very systematic and meticulous manner.
Major comments:
- The negative prognostic impact of double expressor status (DE status, given by coexpression of MYC and BCL2) should be indicated since a significant proportion of samples express both MYC and BCL2.
- The authors have reported not much mutational changes over the time in the primary and relapsed/refractory paired samples in the 52 genes analyzed. Given the fact that the genomic landscape for the hotspot genes did not change in the paired samples over time, how a molecular based approach such as ddPCR will be advantageous for disease monitoring?
Author Response
Reviewer 3
Comments and Suggestions for Authors:
Thank you very much for providing an opportunity to review the article titled “ Genetic Stability of Driver Alterations in Primary Cutaneous Diffuse Large B-cell Lymphoma, Leg Type and Their Relapses a Rationale for the Use of Molecular-Based Methods for More Effective Disease Monitoring” by Anne M.R. Schrader and co-authors.
In this manuscript Anne M.R. Schrader and the co-authors have performed a systematic retrospective investigation on the genetic evolution of primary and relapsed/refractory primary cutaneous diffuse large B-cell lymphoma, leg type (PCDLBCL-LT) using formalin-fixed paraffin-embedded skin biopsies from 57 patients. They have used targeted next-generation sequencing technique to identify mutational status of 52 B-cell lymphoma relevant genes. In addition, patient samples were characterized by using immunohistochemistry and triple FISH for BCL2, MYC and BCL6.
The authors have reported the clinical and immunophenotypic characteristics of these patient samples. The mutational analysis identified several mutations in the key driver genes as well as other genes relevant for the B-cell lymphoma. Overall, majority of the cases had at least one or more main driver mutations of PCDLBCL-LT.
Mutational profiling of the paired samples from 16 primary and relapsed/refractory patient samples also showed at least one hotspot mutation in either MYD88, CD79B or CDKN2A. However, the mutational landscape did not change significantly over time in these paired samples.
Major comments:
The study was conducted in a very systematic and meticulous manner.
Reply: We thank the reviewer for his/her observations.
Major comments:
The negative prognostic impact of double expressor status (DE status, given by coexpression of MYC and BCL2) should be indicated since a significant proportion of samples express both MYC and BCL2.
Reply: As requested, we checked for the influence of DE status on survival of the patients in our cohort but did not find a significant result (p-value 0.25), as corresponds with the results of our previous publication (Schrader et al. AJSP 2018). We added this to the manuscript (Page 9, line 308).
Comment: The authors have reported not much mutational changes over the time in the primary and relapsed/refractory paired samples in the 52 genes analyzed. Given the fact that the genomic landscape for the hotspot genes did not change in the paired samples over time, how a molecular based approach such as ddPCR will be advantageous for disease monitoring?
Reply: As requested, we have changes/added the following sentences in the discussion (Pages 11-12, Line 389-412): “Despite that nearly all PCDLBCL-LT patients in our cohort reached a complete remission after initial therapy (96%), two-thirds still experienced a disease relapse. This high relapse rate suggests that residual disease is not properly detected by our currently standard methods of physical examination and PET-CT scanning [24], implying the need for more accurate assessment of minimal residual disease (MRD) in PCDLBCL-LT patients. The genetic stability of the main driver alterations in PCDLBCL-LT patients, as demonstrated in this study, may serve as a rationale for molecular-based methods of MRD assessment. One such molecular method is analysis of tumour-specific mutations in cell-free tumour DNA obtained from liquid biopsies with digital droplet PCR (ddPCR). This technique is especially suitable for hotspot alterations, such as MYD88 L265P and CD79B Y196, and already allowed for non-invasive real-time monitoring of disease progression in systemic DLBCL patients, as we and others previously described [24-26]. Assessment of the molecular MRD can be important for monitoring disease activity, not only to determine treatment efficacy, but also as an early marker for relapsed disease when a patient is radiologically and clinically in complete remission but the molecular MRD marker is positive. The presence or absence of molecular detectable disease may, subsequently, direct therapeutic strategies, for example by the development of stopping rules for TKI-inhibitors when patients achieved enduring deep molecular remissions. However, these concepts will still need to be investigated in future clinical studies to determine its actual therapeutic value.”
Round 2
Reviewer 1 Report
The authors have addressed all relevant comments in a satisfactory manner
Reviewer 2 Report
The authors replied to all my concerns even for questions that cannot be address due to technical issues.
I think that this revised version is significantly improved.